# Unconventional superconductivity without doping in infinite-layer nickelates under pressure

**Simone Di Cataldo** [1] ✉, **Paul Worm**[1], **Jan M. Tomczak** [1,2], **Liang Si** [3] & **Karsten Held** [1]

High-temperature unconventional superconductivity quite generically emerges from doping a strongly correlated parent compound, often (close to) an antiferromagnetic insulator. The recently developed dynamical vertex approximation is a state-of-the-art technique that has quantitatively predicted the superconducting dome of nickelates. Here, we apply it to study the effect of pressure in the infinite-layer nickelate $Sr_xPr_{1-x}NiO_2$. We reproduce the increase of the critical temperature ($T_c$) under pressure found in experiment up to 12 GPa. According to our results, $T_c$ can be further increased with higher pressures. Even without Sr-doping the parent compound, $PrNiO_2$, will become a high-temperature superconductor thanks to a strongly enhanced self-doping of the Ni $d_{x^2-y^2}$ orbital under pressure. With a maximal $T_c$ of 100 K around 100 GPa, nickelate superconductors can reach that of the best cuprates.

Ever since the discovery of high-temperature superconductivity in $LaBaCuO_4$[1], understanding or even predicting new unconventional (not electron-phonon-mediated) superconductors and identifying the pairing mechanism has been the object of an immense research effort. A new opportunity for a more thorough understanding arose with the discovery of superconductivity in several infinite-layer nickelates $A_{1-x}B_xNiO_2$[2–7], where $A$ = La, Nd, Pr, and $B$ = Sr, Ca are different combinations of rare earths and alkaline-earths. These nickelates are at the same time strikingly similar to cuprates (for this reason theory predicted nickelate superconductivity 20 years before experiment[8]) but also decidedly different. This constitutes an ideal combination to clarify the presumably common mechanism behind superconductivity in both systems. Superconductivity in nickelates was found to be quite independent of the rare earth $A$ and dopant $B$[9] with a dome-like shape characteristic of unconventional superconductors.

Theoretical work has left little doubt that nickelates are, indeed, unconventional superconductors[10–12], and the similarity to the crystal and electronic structure of cuprates is striking[8,13]. There are, however, subtle differences between cuprates and nickelates: Compared to $Cu^{2+}$, the $3d$ bands of $Ni^{1+}$ are separated by a larger energy from the oxygen ones, hence hybridization is weaker and oxygen plays a less prominent role than in cuprates. On the other hand, the rare-earth $A$-derived bands cross the Fermi level in nickelates and form electron pockets. These electron pockets self-dope the Ni $d_{x^2-y^2}$ band with about 5% holes[12,14–20] and prevent the parent compound from being an antiferromagnetic insulator. Given the inherent difficulty in incorporating the effect of strong electronic correlations, different theory groups have arrived at a variety of models for describing nickelates[16,19–26].

Based on a minimal model consisting of a 1-orbital Hubbard model for the Ni $d_{x^2-y^2}$ band[12] plus largely decoupled electron pockets, that only act as electron reservoirs, Kitatani et al.[20] accurately predicted the superconducting dome in Sr-doped $NdNiO_2$[20] prior to experiment[4,4,27]. In particular, the agreement to more recent defect-free films[27] is excellent. This includes the quantitative value of $T_c$, the doping region of the dome, and the skewness of the superconducting dome, see ref. 28. Also pentalayer nickelates[7] seamlessly fit the results of Ref. 28. In Refs. 20,29 some of us pointed out that larger $T_c$'s should be possible if the ratio of interaction to hopping, $U/t$, is reduced.

In a recent seminal paper, Wang et al.[30] reported a substantial increase of $T_c$ in $Sr_xPr_{1-x}NiO_2$ ($x$ = 0.18) films on a $SrTiO_3$ (STO) substrate from 18 to 31 K if a pressure of 12 GPa is applied in a diamond anvil cell. There are no indications of a saturation of the increase of $T_c$

[1]Institut für Festkörperphysik, Technische Universität Wien, 1040 Wien, Austria. [2]King's College London, London WC2R 2LS, UK. [3]School of Physics, Northwest University, Xi'an 710127, China. ✉e-mail: simone.dicataldo@uniroma1.it

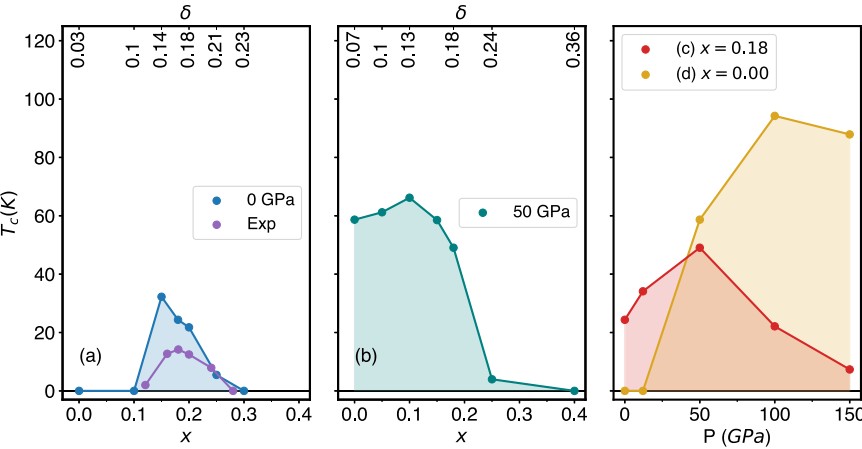

**Fig. 1 | Phase diagram $T_c$ vs. Sr-doping $x$ and pressure $P$ of $Sr_xPr_{1-x}NiO_2$ as calculated in DΓA.** Four different paths are considered: as a function of $x$ for **a** 0 GPa and **b** 50 GPa; as a function of $P$ at **c** $x = 0.18$ and **d** $x = 0$. The secondary (upper) x-axis of panels (**a**), **b** shows the effective hole doping $\delta$ of the Ni $d_{x^2-y^2}$ orbital with respect to half filling. Panel **a** also compares to the experimental result from ref. 58 (purple dots).

with pressure yet. First calculations of the electronic structure, fixing the in-plane lattice constant to the ambient pressure value and relaxing (reducing) the out-of-plane $c$-axis have been presented[31]. A large $T_c$ under pressure[32] or at least a resistivity drop[33], has also been reported in another nickelate: $La_3Ni_2O_7$. With a $3d^{7.5}$ electronic configuration and prevalent charge density wave fluctuations, the mechanism in this compound is, however, clearly distinct from the (slightly doped) $3d^9$ nickelates considered here.

In this work, we employ the same state-of-the-art scheme that was successful in ref. 20, which is based on density functional theory (DFT), dynamical mean-field theory (DMFT), and the dynamical vertex approximation (DΓA). We study the pressure dependence of the superconducting phase diagram of $PrNiO_2$ (PNO) with and without Sr doping. We find (i) a strong increase of the hopping $t$, almost by a factor of two, when going from 0 to 150 GPa, while the value of $U$, obtained through constrained random-phase approximation (cRPA), remains essentially unchanged as in cuprates[34]. Importantly, pressure further results in (ii) deeper electron pockets, effectively increasing the hole doping $\delta$ of the Ni $d_{x^2-y^2}$ band with respect to half-filling. Altogether, this results in the phase diagrams shown in Fig. 1, the main result of our work. When going from (a) ambient pressure to (b) 50 GPa, $T_c$ increases by up to a factor of two and $d$-wave superconductivity is observed in a much wider doping range−quite remarkably even without Sr doping. As a function of pressure, at doping fixed to $x = 0.18$ (c), the simulated phase diagram shows a very similar increase of $T_c$ from 0 to 12 GPa as in experiment[30]. The figure further reveals that $T_c$ will continue to increase up to 49 K at 50 GPa, followed by a rapid decrease at higher pressures. For the parent compound, $PrNiO_2$ ($x = 0$, d), the enhanced self-doping alone is sufficient to turn it superconducting with a maximum predicted $T_c$ of close to 100 K at 100 GPa.

## Results

As the superconducting nickelate films are grown onto an STO substrate, particular care has to be taken when simulating the effect of isotropic pressure in the diamond anvil cell (cf. Supplemental Material (SM) Section IA for a flowchart of the overall calculations and Section IC for further details of the pressure calculation). First, since the thickness of the film is 10−100 nm and thus negligible compared to that of the STO substrate, we calculate the STO equation of state in DFT and, from this, obtain the STO lattice parameters under pressure. Second, we fix the in-plane $a$ (and $b$) lattice parameters to that of STO under pressure and find the lattice parameter $c$ for the nickelate which minimizes the enthalpy at the given pressure. The resulting lattice

constants are shown in Table 1. This procedure reflects the response of the system to the rather isotropic pressures realized in the experiment whereas in ref. 31 the $a$−$b$ lattice parameters had been fixed to that of unpressured STO[31].

With the crystal structure determined, we calculate the DFT electronic structure at pressures of 0, 12, 50, 100, and 150 GPa. Next, we perform a 10- and 1-orbital Wannierization around the Fermi energy, including all Pr-$d$ plus Ni-$d$ orbitals and only the Ni $d_{x^2-y^2}$ orbital, respectively. The DFT band structures and Wannier bands are shown in SM Fig. S5 and as white lines in Fig. 2.

Following the method of refs. 20,35, we supplement the Wannier Hamiltonian with a local intra-orbital Coulomb interaction of $U = 4.4$ eV (2.5 eV) and Hund's exchange $J = 0.65$ eV (0.25 eV) for Ni-3$d$ (Pr-5$d$) as calculated in cRPA in our previous work[36]. For the thus-derived 10-band model, we perform DMFT calculations. The resulting DMFT spectral function of undoped PNO is shown in Fig. 2, for 0 and 50 GPa. We see that the Ni $d_{x^2-y^2}$ orbital crossing the Fermi energy is strongly quasiparticle-renormalized compared to the DFT result (white lines). In addition, there are pockets at the Γ and A momenta, which essentially follow the DFT band structure without renormalization.

Important for the following is that, with the overall increase of bandwidth under pressure, the size of the pockets grows dramatically under pressure in DFT and DMFT alike. The enlargement of the Γ pocket can also be seen from the Fermi surface Fig. 2 (d) vs. (b). The effect at higher pressures is shown in SM Figs. S8 and S9.

The results of the 10-band model show that the low-energy physics of the system boils down to one-strongly correlated Ni $d_{x^2-y^2}$ orbital plus weakly correlated electron pockets. Here, the A-pocket does not hybridize by symmetry with the Ni $d_{x^2-y^2}$ band, as is evident by the mere crossing of both between R and A. The band forming the Γ

**Table 1 | Ab initio values for the lattice constants and the hoppings of the 1-orbital Hubbard model for $PrNiO_2$ under pressure**

| P | a | c | t | t′ | t″ |
|---|---|---|---|---|---|
| [GPa] | [Å] | [Å] | [eV] | [eV] | [eV] |
| 0 | 3.90 | 3.32 | −0.39 | 0.10 | −0.05 |
| 12.1 | 3.83 | 3.20 | −0.42 | 0.10 | −0.05 |
| 50 | 3.67 | 3.03 | −0.48 | 0.11 | −0.06 |
| 100 | 3.54 | 2.89 | −0.56 | 0.11 | −0.07 |
| 150 | 3.45 | 2.79 | −0.62 | 0.12 | −0.07 |

Here, $t$, $t'$, and $t''$ are the nearest, next-nearest, and next-next-nearest neighbor $d_{x^2-y^2}$-hoppings.

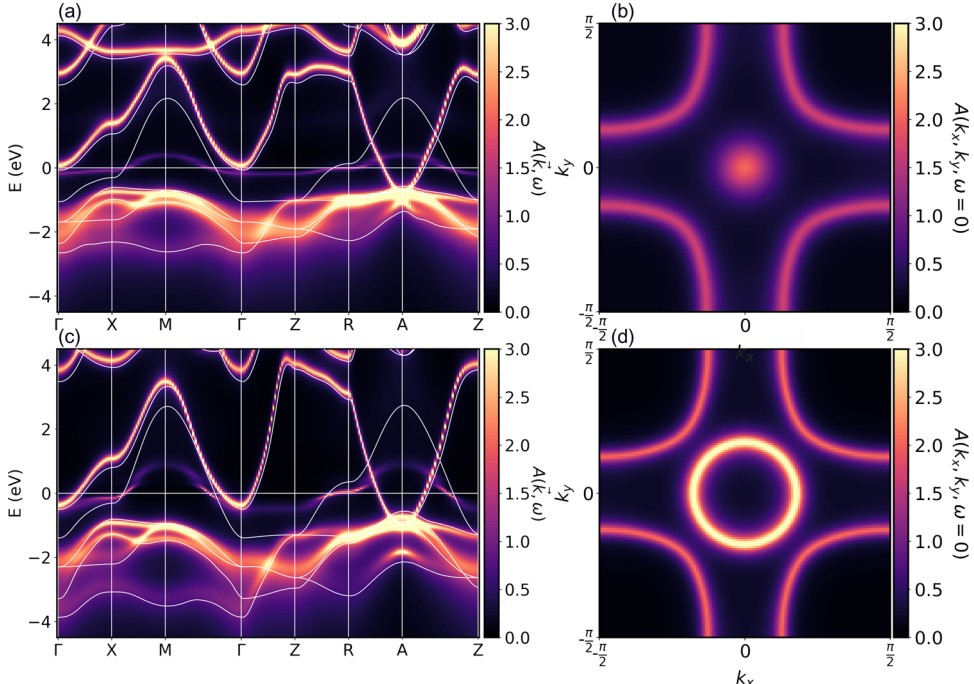

**Fig. 2 | Spectral function for undoped PNO.** Panels **a** and **c** show the DMFT spectral function $A(\mathbf{k}, \omega)$ (color scale) and the Wannier bands (white lines) along a path through the Brillouin zone at temperature $T = 300K$ and a pressure of 0 and 50 GPa, respectively. Panels (**b**) and (**d**) show the same spectral function in the $k_z = 0$ plane. $A(k_x, k_y, k_z = 0, \omega = 0)$. The $\Gamma$ point is at the center, i.e., ($k_x = 0, k_y = 0$).

pocket, on the other hand, mainly hybridizes with other Ni orbitals that, in turn, couple to the Ni $d_{x^2-y^2}$ through Hund's $J$. Note, however, that the main spectral contribution of these other Ni-$d$ orbitals is still well below the Fermi energy in Fig. 2.

The above justifies a one-band minimal model for superconductivity in PNO[13,20,29], with the working hypothesis that superconductivity arises from the correlated Ni $d_{x^2-y^2}$ band only. However, the effective hole-doping $\delta$ of the Ni $d_{x^2-y^2}$ band (relative to half-filling) has to be calculated from the 10-band DFT + DMFT to properly account for the electrons in the pockets. In the following, it is thus imperative to always distinguish between the number of holes corresponding to Sr substitution of the Pr site (chemical doping $x$) and the holes in the Ni $d_{x^2-y^2}$ band compared to half filling (effective doping $\delta$).

The electron pockets induce a nonlinear dependency of $\delta$ from $x$, and their growth with pressure $P$ causes $\delta$ to increase by about 0.06 from 0 to 100 GPa, see SM Fig. S7.

Using the effective doping $\delta$ of the Ni $d_{x^2-y^2}$ band, we perform a second DMFT calculation for the single Ni $d_{x^2-y^2}$ orbital, which we describe as a single-band Hubbard model with an interaction of $U = 3.4$ eV. This $U$ is smaller than for the 10-band model due to additional screening, but it is notably insensitive to pressure (cf. SM Table S2). The main effect of pressure is instead the increase of $t$ as summarized in Table 1 and the already mentioned enhanced self-doping.

In Fig. 3, we show the spectral function $A(\mathbf{k}, \omega)$ of the 1-band model for PNO as a function of pressure and $x = 0$. The panels for 0 and 50 GPa can be compared to Fig. 2a, c and show that the 1-band model reproduces the renormalization of the Ni $d_{x^2-y^2}$ orbital in the fully-fledged 10-band calculation (see Table S4) for a quantitative comparison.

Hubbard bands are visible at all pressures in Fig. 3, but become more spread out and less defined with increasing pressure. Simultaneously, the effective mass $m^*$ decreases, and the bandwidth widens. Similar results but for the experimentally investigated Sr-doping $x = 0.18$ can be found in SM Fig. S12 and as a function of doping at 50 GPa in Fig. S11.

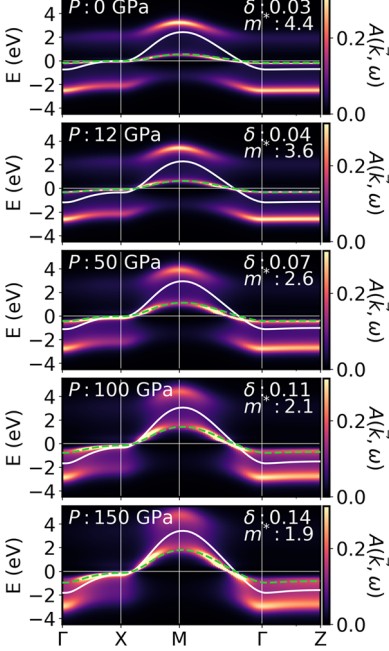

**Fig. 3 | DMFT spectral function $A(k_x, k_y, k_z = 0, \omega)$ (color bar) for undoped PNO for the 1-band model and as a function of pressure from 0 to 150 GPa at $T = 0.0125t$.** The original DFT Wannier band and the same band renormalized with the effective mass are shown as solid white and dashed green lines, respectively.

Next, we calculate the superconducting $T_c$ using DΓA[35,37]. In Fig. 1, we follow four different paths in parameter space: as a function of doping at (a) 0 GPa and (b) 50 GPa as well as as a function of pressure with Sr-doping (c) $x = 0.18$, i.e., for the parent compound, and (d) $x = 0$. At ambient pressure in Fig. 1a our results are in excellent agreement with our previous calculations for other nickelates[20,29], cf. SM Fig. S14.

The small differences are ascribable to the slightly different material and are in good agreement with experiment[3,27]. We find the effect of pressure to be significant: At 50 GPa, the maximum $T_c$ is enhanced by a factor of two compared to ambient pressure, while the maximum slightly shifts to lower doping ($x = 0.10$). Remarkably, even the undoped parent compound becomes superconducting at 50 GPa [$x = 0$ in Fig. 1b], due to the increased self-doping from the electron pockets.

The experimental Sr-doping of $x = 0.18$[38] is close to optimal doping at 0 GPa. Increasing pressure in Fig. 1c, we observe an increase of $T_c$ by 0.81 K/GPa in excellent agreement with the experimental rate of 0.96 K/GPa[38], for pressures up to 12 GPa. The predicted $T_c$ of 30 K at 0 GPa is slightly higher in theory than the experimental 18 K[38], but still in good agreement. As pressure increases beyond 12 GPa, $T_c$ continues to grow and peaks at around 50 GPa with 49 K, before decreasing for higher pressures.

Most striking is the result for the undoped compound PNO in Fig. 1d. Here, superconductivity sets in below 50 GPa and peaks at almost 100 K around 100 GPa. Intrinsic doping from the electron pockets is sufficient to make the parent compound superconducting at high temperatures.

## Discussion

To rationalize our results, we plot the four paths at a fixed pressure, respectively, fixed Sr-doping $x$ in Fig. 4, but now as a function of $U/t$ and the effective hole doping $\delta$ of the Ni $d_{x^2-y^2}$ orbital. Superimposed is the DΓA superconducting eigenvalue $\lambda$ as calculated already in[29], which qualitatively agrees with dynamical cluster (DCA) calculations performed at higher $T$'s[39]. The darker gray regions correspond to a higher $T_c$. At small dopings we show no superconducting eigenvalue as it competes here with antiferromagnetic order, see, e.g.,[40] and the Supplemental Material.

The application of an isotropic pressure on infinite-layer PrNiO$_2$ has two effects: First, it boosts the hopping $t$ of the Ni $d_{x^2-y^2}$ orbital, which at 150 GPa becomes almost twice as large as at 0 GPa, see Table 1. This increases the overall energy scale and thus enhances $T_c$. Since $U$ does not change significantly, the ratio $U/t$ also decreases. This is preferable for superconductivity since at ambient pressure PNO exhibits a $U/t$ slightly above the optimum of $U/t \sim 6$ (above the darker gray region in Fig. 4). However, at high pressures of e.g., 100 GPa and 150 GPa curves (c) and (d) have passed the optimum in Fig. 4; $T_c$ in Fig. 1 decreases again.

Second, pressure enhances the effective hole doping $\delta$ even at fixed Sr-doping $x$, as the electron pockets become larger. For this reason, curves (c) and (d) in Fig. 4 deviate from a vertical line. For Sr-doping (c) $x = 0.18$, which is close to optimum at 0 GPa, the curve moves away from optimum doping to the overdoped region when pressure is applied. This is a major driver for the decrease of $T_c$ above 50 GPa in Fig. 1c. In stark contrast, for the parent compound ($x = 0$; d) the effective hole doping $\delta$ goes from underdoping to optimal doping, and only at much larger pressures to overdoping and too small $U/t$. Consequently, PNO without doping hits the sweet spot for superconductivity in Fig. 4 at a pressure between 50 and 100 GPa.

In short, our results strongly suggest that experiments for infinite-layer nickelates are still far from having achieved their maximum $T_c$. Surprisingly, the maximum $T_c$ of almost 100 K is predicted to be found in undoped PrNiO$_2$ between 50 and 100 GPa. This places nickelates almost on par with cuprates in the Olympus of high-$T_c$ superconductors. The nickelate phase diagram under pressure will not only exhibit a significant increase in $T_c$ but also a wider dome. In particular, the maximum of this dome is shifted to lower Sr-doping $x$ when a pressure of 50–100 GPa is applied.

Such pressures can be achieved experimentally in diamond anvil cells. An alternative route to obtain the same in-plane lattice compression is using a substrate with smaller lattice parameters. For example, LaAlO$_3$, YAlO$_3$, and LuAlO$_3$ have lattice parameters of 3.788[41], 3.722[42], and 3.690 Å[43], respectively, which are already close to the in-plane lattice constants at 50 GPa. As this approach would not change the out-of-plane lattice parameter, the self-doping of the Ni $d_{x^2-y^2}$ band from the electron pockets should be less important. Hence, we expect that with these substrates a higher $T_c$ might be achieved, but only with at least 10% doping.

## Methods

In this section, we summarize the computational methods employed. The interested reader can find additional information in Supplementary Material, and data and input files for the whole set of calculations in the associated data repository[44].

Density functional theory calculations were performed using the Vienna ab initio simulation package (VASP)[45,46] using projector-augmented wave pseudopotentials and Perdew-Burke-Ernzerhof exchange-correlation functional adapted for solids (PBESol)[47,48], with a cutoff of 500 eV for the plane wave expansion. Integration over the Brillouin zone was performed over a grid with a uniform spacing of 0.25 Å$^{-1}$ and a Gaussian smearing of 0.05 eV. Wannierization was performed using `wannier90`[49].

The Hubbard $U$ of the 1-orbital setup under pressure was computed from first principles using the constrained random phase approximation (cRPA) in the Wannier basis[50] for entangled band-structures[51] relying on a DFT electronic structure obtained from a full-potential linearized muffin-tin orbital method[52].

DMFT calculations were performed using `w2dynamics`[53], with values of $U, J,$ and $t$ as detailed in the main text and in SM Tables S1 and S3. The 10-band calculations were performed at a temperature of 300 K, with a total of 30 iterations to converge the local Green's function, and a final step with higher sampling. The 1-band DMFT calculations were performed at variable temperatures, with a total of 70 iterations, and a final step with higher sampling, which was increased at lower temperatures. Self-consistency between DMFT and DFT was not considered.

The calculation of the non-local quantities via ladder DΓA is done as before[20] as follows: (1) Starting from the local irreducible particle-hole vertex obtained with `w2dynamics`[54], we calculate non-local susceptibilities and vertices using the `DGApy` code[55]. Here, we calculate the particle-hole ladder and use the Moriya $\lambda$ correction to mimic self-consistency. (2) The non-local vertices of (1) are fed into the particle-particle channel solving the linearized Eliashberg equation. This is akin

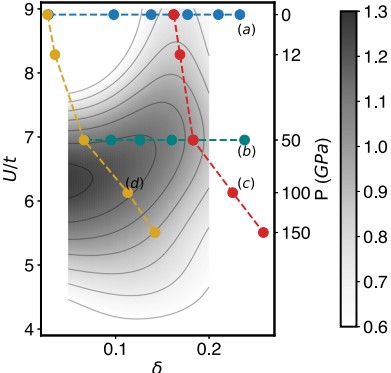

**Fig. 4 | Strength of superconductivity with respect to band filling and $U/t$.** The considered four paths in the $U/t$ vs. effective hole doping $\delta$ parameter space are: at a fixed pressure of **a** 0 GPa and **b** 50 GPa as a function of Sr-doping $x$; as a function of pressure for fixed Sr-doping **c** $x = 0.18$ and **d** $x = 0.00$ (blue, green, red, and yellow dots, respectively). The gray color bar and the isocontour lines indicate the strength of superconductivity (superconducting eigenvalue $\lambda$ at $T = 0.01t$; data from Fig. 3 of ref.[29]). The secondary $y$-axis reports the pressure corresponding to the $U/t$ values shown.

to one parquet step without self-consistency (which would require feeding back the particle–particle corrections to the particle–hole channel). See Supplemental Material for further details and discussions.

## Data availability

All the data generated in this study, along with input and output files are available in a repository hosted by TU Wien (see[44]). Related data, including the local Green's functions and DFT calculations are also available in the `NOMAD` repository[56,57].

## Code availability

`Quantum ESPRESSO` and `w2dynamics` are publicly available codes. The `DGApy` code employed in this work was also made publicly available at:[55].

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

## Acknowledgements

We would like to thank Motoharu Kitatani and Juraj Krsnik for the helpful discussion. We further acknowledge funding through the Austrian Science Funds (FWF) projects ID I 5398, I 5868, P 36213, SFB Q-M&S (FWF project ID F86), and Research Unit QUAST by the Deutsche Forschungsgemeinschaft (DFG; project ID FOR5249). L.S. is thankful for the starting funds from Northwest University. Calculations have been done on the Vienna Scientific Cluster (VSC). This research was funded in whole or in part by the Austrian Science Fund (FWF) [10.55776/I5398]. For open access purposes, the authors have applied a CC BY public copyright license to any author accepted manuscript version arising from this submission.

## Author contributions

S.D.C. performed the DFT and DMFT calculations. S.D.C. and P.W. performed the DΓA calculations. J.M.T. performed the cRPA calculations. L.S. and K.H. designed and supervised the project. All authors participated in the discussion, contributed to the writing of the paper, and approved the submitted version.

## Competing interests

The authors declare no competing interests.
