## [Peer Review File · Nature Communications]

Unconventional superconductivity without doping in infinite-layer nickelates under pressureREVIEWER COMMENTS

Reviewer #1 (Remarks to the Author):

In this article, the authors consider the pressure and doping dependence of superconductivity in the family of nickelates $\text{Sr}_x\text{Pr}_{1-x}\text{NiO}_2$. Superconductivity in this class of materials is a much discussed topic, following the seminal discovery in the group of Harold Hwang. Here, the authors apply a theoretical machinery that has proven highly successful for these systems, namely the DGammaA extension of dynamical mean-field theory in combination with state of the art electronic structure. They predict that the Praseodymium-based material have the ability to reach a rather high $T_c \sim 100\text{K}$ under pressure, with enhanced self-doping of the Ni d-orbital. I find the methodology convincing and the results of great interest. I therefore recommend publication in Nature Communications without further changes.

Reviewer #2 (Remarks to the Author):

In the paper entitled “Unconventional superconductivity without doping: infinite-layer nickelates under pressure”, the authors study the interplay of doping and pressure on superconducting transition temperature, T_c in the infinite layer nickelate superconductor, strontium doped PrNiO. This topic is of considerable current interest. Their key findings are: increase of T_c with pressure as seen in experiments, and that under sufficiently large pressure, the undoped compound PrNiO₂ would go superconducting with a high transition temperature. The results are summarized nicely in phase diagrams showing T_c vs doping at fixed pressures, and vs pressure at fixed doping.

The results presented are based on a sizeable amount of computational work. This is commendable. According to the workflow provided in the Supplementary Information, wannierization of DFT based electronic structure leads to a 10-band model on which they perform DMFT calculations, and then a second DMFT on a 1-band model. The value of correlation U is obtained from a separate cRPA calculation and input into the DMFT calculations. Wannierization allows them to also obtain hopping parameters. The doped compounds are simulated using VCA, and checked for consistency for one case, viz. $x = .25$ with a $2 \times 2 \times 2$ supercell calculation. To obtain T_c , they employ ladder-DGammaA and Eliashberg equation on the 1-band model obtained from their DFT + DMFT calculations.

The work presented here follows previous work by a subset of authors from the same group, using methods as used here. Their previous papers (Refs. 13, 20 of this paper) presented phase diagrams of superconducting T_c as a function of doping in the doped nickelates. Thus, the current paper may be considered an addition, albeit useful, in that this focuses more on T_c as function of pressure at fixed doping. Effects of the interplay of doping and pressure on the nickelate superconductors has been studied previously; e.g. Ref. 32 of this paper, though that work did not calculate T_c .

In my view, several aspects of the paper, however, need to be addressed by the authors:

i) In obtaining T_c , the authors carry out ladder DGammaA and linearized Eliashberg equation on a 1-

band model. Local particle-hole irreducible vertex is obtained from the local (DMFT) 2-particle Green's fn. To obtain the χ'' lattice susceptibility, the authors use the so-called Moriya λ -correction, presumably to correct issues with high-frequency behavior. However, it is not clear if this is sufficient or convincing enough in obtaining the desired susceptibility.

ii) Extrapolation to $\lambda \rightarrow 1$ is done using an expression given in Sup. Inf. G. Even though this is based on Ref. 2 of this paper, the authors should clearly define and explain the quantities involved, namely, of A , B , n , p , β , as well as the degree to which this works at a satisfactory level.

iii) The ladder χ'' calculation is not done in a self-consistent manner; for example, the feedback of the particle-particle fluctuations of the p-h fluctuation are not considered. Admittedly, this is a computationally expensive task, nevertheless, the authors should clearly point this out in the main text, and comment on possible ways their results would be modified as a consequence.

iv) As pointed out in Ref 29 of this paper, there may be a divergence of the irreducible vertex itself not connected to the superconducting phase transition. Then one has to be careful in not taking the highest eigenvalue λ , rather eigenvalues close to $\lambda \rightarrow 1$. According to Sup. Inf. G, the authors appear to examine $\lambda \rightarrow 1$. It is not clear, and hence should be pointed out in the main text if this was done because of some divergence of the irreducible vertex or other reasons.

v) The DMFT calculations are done at a single $T = 300\text{K}$. It would be useful for the reader to know of possible effects on superconducting transition temperature calculations if DMFT calculations were to be done at lower T 's. Also the author should comment on the reason for the choice of this T and if they are able to do calculations at lower T 's.

vi) The DMFT calculation are done on downfolded models and single-shot (i.e no self-consistency is enforced between the DFT and DMFT). It would be useful for the readers to be aware of this, and so the authors should point this out in the main text.

Overall, the paper constitutes an important contribution in the area of nickelate superconductivity, and will advance understanding of these interesting materials.

In view of this work being more of the incremental kind, and that there has been previous study on the interplay of doping and pressure, it is rather difficult to view this work as truly novel and hence recommend publication in Nature Communications; it may be better suited as a Scientific Report or perhaps as a Communication Physics. But in any case, the points raised above, to some extent, diminishes the astuteness of the results at a quantitative level, and need to be addressed prior to publication in any format.

Reviewer #3 (Remarks to the Author):

Report for Cataldo et al. (nature comm) -- round 1

The authors of the manuscript "Unconventional Superconductivity without Doping: Infinite-Layer Nickelates under Pressure" present an in-depth ab-initio study of the superconducting properties of infinite-layer nickelate compounds using DFT+DMFT+DGA calculations. The manuscript is well-

organized and discusses new discoveries in a currently trending topic, particularly after the discovery of superconductivity in the 327 LNO compound under pressure.

The authors start with carefully extracting low-energy models under pressure using DFT calculations. This process is well-detailed and accurate, though most of the specifics are in the supplementary materials. Using the virtual crystal approximation and performing detailed equation of state analysis they simulate the in-plane pressure from the diamond anvil cell in experiment.

Next, DFT+DMFT calculations are performed with interaction parameters calculated via cRPA. Finally, the superconductivity transition temperatures are determined by solving linearized Eliashberg equations at different temperatures. The interaction vertices are computed using the DGA method. It is found that the critical temperature for superconductivity can be significantly improved in these compounds when applying pressure by pushing the material to a more favorable U/t ratio.

Once my following questions are addressed, I will be more than happy to recommend the manuscript for publication:

1. At certain points, the manuscript's language adds unnecessary adjectives or it diminishes other work where it's not necessary. For example:

- "In this work, we employ the same state-of-the-art scheme that was *so* successful ... "
- "This procedure better reflects the response of the system to the rather isotropic pressures realized in experiment and is more realistic than that used in Ref. [32] where the a-b lattice parameters had been fixed to that of unpressured STO [32]."

I think the authors can shorten these sentences. The results of their work speak by themselves.

2. Fig 2: The top row and bottom row appear to show different pressures. This point is not mentioned in the caption.

3. Fig 4: The process for obtaining the gray-shaded highlighting of the leading eigenvalue is unclear. Are there additional calculations performed beyond those indicated by the colored circles? If so, it would be helpful to mention this.

4. Do the one-band DMFT calculations utilize the full $H(R)$, or only the three hoppings specified in the table? Could the authors also provide a quantification of how closely the mass-enhancements align between the one-band and ten-band models? The text briefly mentions that they agree well, but offers no further detail.

5. Why didn't the authors include Oxygen p Wannier functions in the large energy window calculation? This could affect how the Coulomb interaction is screened in the cRPA procedure mentioned in the manuscript. Were the Oxygen bands factored into the disentanglement to accurately capture this? I appreciate the effort of uploading the full data set. However, I cannot access it (protected by institute login)

6. The extraction of Kanamori parameters from the U_{ijkl} tensor from cRPA is not explained. Are these parameters extracted from the static limit ($w=0$) and then symmetrized? How well does the fit work?

7. I'm uncertain why the authors chose to use the Kanamori Hamiltonian in a full 5 orbital calculation for the wide energy DMFT calculations. This choice is inappropriate for a full 5 orbital d-shell due to the symmetry of the interaction. Why did the authors not opt for a Slater Hamiltonian instead? This could be a significant approximation, and the authors should at least acknowledge this fact.

8. I'm unclear as to why the bare Coulomb interaction remains unchanged under pressure in the one orbital cRPA calculation. The band structures show a significant change in the bandwidth of the wannierized dx_2y_2 orbital. How does this not affect the bare Coulomb interaction? Is the spread of the Wannier functions constant under pressure? If so, do the authors have a theory as to why this is the case?

9. The authors discuss the AFM order rather vaguely. As demonstrated by cluster DMFT studies (and other single site DMFT studies) found at <https://doi.org/10.1103/PhysRevB.105.205131>, the AFM order competes with superconductivity at around $\delta=0.12$ (filling of dx_2y_2 orbital) in these compounds. This competition could be due to the mean field nature of the calculations. Can the authors elaborate on how or if such a state could be observed in DGA? Is it correct to assume that the AFM order is essentially suppressed, or did the authors not observe any AFM order in their DMFT calculations? Is it known how the AFM phase evolves under pressure in simple one-band models and competes with the superconductivity?

Let us start by thanking the referees for the careful reading of our manuscript and their positive comments. Please find below a point-by-point rebuttal.

Simone Di Cataldo, Paul Worm, Jan Tomczak, Liang Si and Karsten Held

REPLY TO REVIEWER #1

Reviewer #1 (Remarks to the Author): In this article, the authors consider the pressure and doping dependence of superconductivity in the family of nickelates $\text{Sr}_x\text{Pr}_{1-x}\text{NiO}_2$. Superconductivity in this class of materials is a much discussed topic, following the seminal discovery in the group of Harold Hwang. Here, the authors apply a theoretical machinery that has proven highly successful for these systems, namely the DGammaA extension of dynamical mean-field theory in combination with state of the art electronic structure. They predict that the Praseodymium-based material have the ability to reach a rather high $T_c \sim 100\text{K}$ under pressure, with enhanced self-doping of the Ni d-orbital. I find the methodology convincing and the results of great interest. I therefore recommend publication in Nature Communications without further changes.

We thank the referee for the enthusiastic comments, appraising the results, acknowledging that they are of "great interest", and the unequivocally positive recommendation.

REPLY TO REVIEWER #2

Reviewer #2 (Remarks to the Author): In the paper entitled “Unconventional superconductivity without doping: infinite-layer nickelates under pressure”, the authors study the interplay of doping and pressure on superconducting transition temperature, T_c in the infinite layer nickelate superconductor, strontium doped PrNiO. This topic is of considerable current interest. Their key findings are: increase of T_c with pressure as seen in experiments, and that under sufficiently large pressure, the undoped compound PrNiO₂ would go superconducting with a high transition temperature. The results are summarized nicely in phase diagrams showing T_c vs doping at fixed pressures, and vs pressure at fixed doping. The results presented are based on a sizeable amount of computational work. This is commendable. According to the workflow provided in the Supplementary Information, wannierization of DFT based electronic structure leads to a 10-band model on which they perform DMFT calculations, and then a second DMFT on a 1-band model. The value of correlation U is obtained from a separate cRPA calculation and input into the DMFT calculations. Wannierization allows them to also obtain hopping parameters. The doped compounds are simulated using VCA, and checked for consistency for one case, viz. $x = .25$ with a $2 \times 2 \times 2$ supercell calculation. To obtain T_c , they employ ladder-DG γ A and Eliashberg equation on the 1-band model obtained from their DFT + DMFT calculations. The work presented here follows previous work by a subset of authors from the same group, using methods as used here. Their previous papers (Refs. 13, 20 of this paper) presented phase diagrams of superconducting T_c as a function of doping in the doped nickelates. Thus, the current paper may be considered an addition, albeit useful, in that this focuses more on T_c as function of pressure at fixed doping.

First of all, we would like to thank the referee for the careful reading, the efforts and time spent on the report, as well as for several very helpful technical points which we address below. The main criticism of the referee is that they consider our work a follow-up to our previous calculations of the phase diagram without pressure. With all due respect, whether this is a follow-up or an important insight, depends on the results. And, what we find is indeed unprecedented: unconventional superconductivity at very high temperatures without any chemical doping. While this is a theoretical calculation, our successful prediction of the phase diagram at ambient pressure and the reproduction of the T_c increase in previous experiments (at pressures of up to 12 GPa) gives us confidence that our current prediction is correct. It can and will be checked experimentally. If confirmed, we would expect the experiment to be rather under review in Nature than in Nature Comm.

Effects of the interplay of doping and pressure on the nickelate superconductors has been studied previously; e.g. Ref. 32 of this paper, though that work did not calculate T_c .

We would like to remark that the way in which pressure is studied in Ref. 32 and our work is fundamentally different. In Ref. 32, Christiansson, Petocchi and Werner studied the effect of a reduction of the c axis, while keeping the a and b axis fixed, which models a strain in only one direction. First, this is an unphysical setup, even for uniaxial pressure experiments. Indeed any material’s Poisson ratio is finite: the compression in c -direction will cause the in-plane lattice constant to change. Second, in high-pressure experiments using diamond anvil cells, the pressure is essentially hydrostatic/isotropic, and the a/b compression cannot be neglected, especially above a few tens of GPa. Albeit similar at first glance, our study differs deeply from Ref. 32 in that we study completely different physical conditions.

With the electronic structure being very sensitive to structural changes, seemingly minor details can make a profound difference. This is why, in our work, we went to great length to perform a proper treatment of isotropic pressure. As detailed in the Supplementary Material and as is appreciated by Reviewer #3: (1) We calculated STO under isotropic pressure. (2) As the STO substrate is much thicker, this will yield the a, b in-plane lattice constant to a very good approximation also for the nickelate film as long as the film does not develop faults. (3) We then calculated the nickelate c -axis under isotropic pressure and fixed in-plane lattice constant.

In my view, several aspects of the paper, however, need to be addressed by the authors: i) In obtaining T_c , the authors carry out ladder DG γ A and linearized Eliashberg equation on a 1-band model. Local particle-hole irreducible vertex is obtained from the local (DMFT) 2-particle Green’s fn. To obtain the DG γ A lattice susceptibility, the authors use the so-called Moriya lambda-correction, presumably to correct issues with high-frequency behavior. However, it is not clear if this is sufficient or convincing enough in obtaining the desired susceptibility.

The Referee is referring here to the magnetic susceptibility. The latter and the k -dependent self-energy have been shown to be in excellent agreement with other numerical methods such as diagrammatic Monte Carlo for prototypical many-body models, see Phys. Rev. X 11, 011058 (2021). We now emphasize this in the Supplemental Material, Sect. IG.

ii) Extrapolation to $\lambda \rightarrow 1$ is done using an expression given in Sup. Inf. G Even though this is based on Ref. 2 of this paper, the authors should clearly define and explain the quantities involved, namely, of A, B, n, p, β , as well as the degree to which this works at a satisfactory level.

We thank the referee for pointing this out. We improved the presentation in the Supplemental Information. That is, we now introduce all fit parameters (“np” just stood for numpy and has been removed). Further, we now show in Fig. S13 that even when the lowest two temperature points are not included in the fit, the function is able to extrapolate the trend well, with an error due to extrapolation of $\pm 10K$ in this worst-case scenario.

iii) The ladder DGammaA calculation is not done in a self-consistent manner; for example, the feedback of the particle-particle fluctuations of the p-h fluctuation are not considered. Admittedly, this is a computationally expensive task, nevertheless, the authors should clearly point this out in the main text, and comment on possible ways their results would be modified as a consequence.

That is a valid point of the referee. We now make this more clear in the manuscript. Feeding back the particle-particle fluctuations to the particle-hole fluctuations would require solving the full parquet equations which is not possible at the temperatures studied. As for possible modifications of our results: It is clear that there will be only a feedback effect in the immediate vicinity of the superconducting transition, since only here the particle-particle fluctuations can majorly effect the particle-hole channel. There, the feedback will somewhat dampen (screen) the spin fluctuations. Whether this increases or decreases T_c is difficult to predict, but we expect this effect to be small, as soon as Kosterlitz-Thouless physics is suppressed (e.g., due to a minimal coupling in the third dimension or finite extension in-plane that actually make mono-layer cuprates superconducting, see Nature 575, 156 and J.Phys.A: Math.Theor. 54, 315001).

Let us further emphasize that the dynamical cluster approximation (DCA) yields qualitatively a very similar superconducting eigenvalue λ pattern as DGA in Fig. 4 (see PRL 115, 116402), but with quantitatively smaller λ because DCA cannot reach temperatures as low as ladder DGA.

iv) As pointed out in Ref 29 of this paper, there may be a divergence of the irreducible vertex itself not connected to the superconducting phase transition. Then one has to be careful in not taking the highest eigenvalue lambda, rather eigenvalues close to lambda \rightarrow 1. According to Sup. Inf. G, the authors appear to examine lambda \rightarrow 1. It is not clear, and hence should be pointed out in the main text if this was done because of some divergence of the irreducible vertex or other reasons.

In the current case, because of the doping, there is no vertex divergence, and λ is indeed the leading (largest) eigenvalue. That it grows quite substantially with decreasing temperatures is due to the superconducting phase transition. We now mention this in the Supplemental Material, Sect. II B.

v) The DMFT calculations are done at a single $T = 300\text{K}$. It would be useful for the reader to know of possible effects on superconducting transition temperature calculations if DMFT calculations were to be done at lower T 's. Also the author should comment on the reason for the choice of this T and if they are able to do calculations at lower T 's.

Figure 1: Analytically continued local Green's function for the five orbitals of Ni (left) and Pr (right) in undoped PrNiO₂ at 50 GPa, 100 K (top) and 300 K (bottom). The correlated spectral functions for the d_{z^2} , d_{xz} , d_{yz} , $d_{x^2-y^2}$, and d_{xy} orbitals are shown as solid blue, orange, green, red and purple lines, respectively. The uncorrelated, DFT DOS for Ni $d_{x^2-y^2}$ is shown as a black dashed line (top left panel).

The Reviewer here refers to the 10-bands DMFT calculations employed to determine the occupation of the Ni- $d_{x^2-y^2}$ orbital, while the 1-band calculations were done at different temperatures. We did the DMFT calculation at room temperature and did not do calculations at further temperatures because at a temperature well below the quasiparticle bandwidth, we did not expect differences on the DMFT level.

To make sure that we did not overlook something here, we now performed additional 10-bands calculations at lower temperatures, for 50 GPa at 100K, in the absence of doping. In Figs. 1 and 2 we compare the spectrum and self energy, respectively, for 50 GPa at 300K to that at 100K. Indeed both are almost the same. Also the relevant occupation of the Ni $d_{x^2-y^2}$ orbital changes by less than 10^{-4} per Ni. This confirms that temperature variations do not impact the DMFT results. We discuss the new lower- T calculations in the Supplemental Material.

Figure 2: Local selfenergy for the Ni and Pr $d_{x^2-y^2}$ orbital at 100 and 300 K (solid blue and dashed orange lines, respectively). **Note:** the data were plotted skipping one in two data points for visual clarity.

vi) The DMFT calculation are done on downfolded models and single-shot (i.e no self-consistency is enforced between the DFT and DMFT). It would be useful for the readers to be aware of this, and so the authors should point this out in the main text.

As an additional check to assess the possible influence of self-consistency in the charge between the DFT and DMFT cycle, we compared the occupation of the most strongly correlated orbital, i.e. Ni $d_{x^2-y^2}$ at 50 GPa (See Fig. 1). We obtain an occupation of this orbital of 0.41 in the uncorrelated case and 0.49 in the correlated one. The occupation of all the other orbitals changes even less. Given the small difference in occupations we expect the effect of self-consistency between DFT and DMFT to be minor. We now mention this in the paper's Supplemental Material, Sect. IF, and briefly in the Methods section of the main text.

Overall, the paper constitutes an important contribution in the area of nickelate superconductivity, and will advance understanding of these interesting materials. In view of this work being more of the incremental kind, and that there has been previous study on the interplay of doping and pressure, it is rather difficult to view this work as truly novel and hence recommend publication in Nature Communications; it may be better suited as a Scientific Report or perhaps as a Communication Physics. But in any case, the points raised above, to some extent, diminishes the astuteness of the results at a quantitative level, and need to be addressed prior to publication in any format.

We thank the referee for acknowledging the "important contribution" of our paper, but we strongly disagree with the second sentence. The fact that the technique of DFT+DMFT+DΓA was employed for other nickelates previously cannot be used to imply that this work is incremental. Realistic calculations for the isotropic pressure studied in experiment have not been performed before for infinite-layer nickelates. Our paper is not a methodology paper or a first application of our method to a new material. Rather, it is the application of an established method in a realistic setting, to offer an exciting *ab initio* prediction of the properties of a material. Our results reveal exciting physics: unconventional superconductivity without chemical doping is possible and rather straightforward to realize experimentally.

REPLY TO REVIEWER #3

Reviewer #3 (Remarks to the Author): The authors of the manuscript "Unconventional Superconductivity without Doping: Infinite-Layer Nickelates under Pressure" present an in-depth ab-initio study of the superconducting properties of infinite-layer nickelate compounds using DFT+DMFT+DGA calculations. The manuscript is well-organized and discusses new discoveries in a currently trending topic, particularly after the discovery of superconductivity in the 327 LNO compound under pressure.

The authors start with carefully extracting low-energy models under pressure using DFT calculations. This process is well-detailed and accurate, though most of the specifics are in the supplementary materials. Using the virtual crystal approximation and performing detailed equation of state analysis they simulate the in-plane pressure from the diamond anvil cell in experiment. Next, DFT+DMFT calculations are performed with interaction parameters calculated via cRPA. Finally, the superconductivity transition temperatures are determined by solving linearized Eliashberg equations at different temperatures. The interaction vertices are computed using the DGA method. It is found that the critical temperature for superconductivity can be significantly improved in these compounds when applying pressure by pushing the material to a more favorable U/t ratio.

Once my following questions are addressed, I will be more than happy to recommend the manuscript for publication:

We thank the Reviewer for appreciating our calculations and results, for the positive recommendation, and, in particular, for spotting a few flaws that needed mending.

1. At certain points, the manuscript's language adds unnecessary adjectives or it diminishes other work where it's not necessary. For example: - "In this work, we employ the same state-of-the-art scheme that was *so* successful ... " - "This procedure better reflects the response of the system to the rather isotropic pressures realized in experiment and is more realistic than that used in Ref. [32] where the a - b lattice parameters had been fixed to that of unpressured STO [32]." I think the authors can shorten these sentences. The results of their work speak by themselves.

That point is well taken. We have edited the manuscript accordingly.

2. Fig 2: The top row and bottom row appear to show different pressures. This point is not mentioned in the caption.

We warmly thank the referee for noticing this missing piece of information! We fixed the caption.

3. Fig 4: The process for obtaining the gray-shaded highlighting of the leading eigenvalue is unclear. Are there additional calculations performed beyond those indicated by the colored circles? If so, it would be helpful to mention this.

We realize the caption was not clear enough. We have tried to make it clearer now. The gray-shaded highlighting was obtained using data from Fig. 3 of Ref 30 (PRL 130, 166002 (2023)).

4. Do the one-band DMFT calculations utilize the full $H(R)$, or only the three hoppings specified in the table? Could the authors also provide a quantification of how closely the mass-enhancements align between the one-band and ten-band models? The text briefly mentions that they agree well, but offers no further detail.

The one-band calculations only use the three hoppings. We now realize this should be made clearer. The procedure for the one-band model is to take the Wannier Hamiltonian $H(\vec{R})$, set to zero any components other than t , t' , and t'' , and then doing the Fourier transform. We clarified this aspect in the Supplementary Information, Sect. ID. For what concerns the quantitative comparison between the 10- and 1-band model, indeed a quantitative comparison was missing. In Tab. S4 we now compare the mass enhancement for the two cases as a function of pressure (doping $x = 0$), extracted directly from the real part of the self-energy along the imaginary axis.

5. Why didn't the authors include Oxygen p Wannier functions in the large energy window calculation? This could affect how the Coulomb interaction is screened in the cRPA procedure mentioned in the manuscript. Were the Oxygen bands factored into the disentanglement to accurately capture this? I appreciate the effort of uploading the full data set. However, I cannot access it (protected by institute login)

In the cRPA, DMFT, and the Wannier functions, we consistently restricted ourselves to the 5 "Ni" and 5 "Pr d -orbitals". These orbitals have quite some oxygen admixture and require somewhat smaller Coulomb interactions because of the screening of the bands of predominant oxygen p character. This procedure is, while formally less complete, somewhat more reliable and less volatile when doing parameter-free calculations than similar calculations including the oxygen orbitals. The reason for this is that (i) the oxygen p -bands are too close to the Fermi energy in DFT and (ii) the p - d splitting is sensitive to the DFT+DMFT double counting scheme used. There have been several attempts to overcome this, including the so-called exact exchange, self-energy self-consistency and using self-interaction corrected (SIC)-DFT. But in our view the more complete calculation including "oxygen p " bands is not better as long as these predominately oxygen bands remain fully filled.

The disentanglement energy window was set from roughly -5 to +15 eV around the Fermi energy (details depend on the specific pressure), while the frozen energy window was roughly from -2 to +3 eV around the Fermi energy (hence factoring in the mainly Ni- d bands). The disentanglement was performed until the Gauge-invariant part of the spread was converged within 10^{-10} Å². As an illustrative example, in Fig. 3 we show the real-space Wannier function associated with the Ni $d_{x^2-y^2}$ band, which effectively includes a partial oxygen character in its tails.

Figure 3: Maximally-localized Wannier function in real space corresponding to the the Ni - $d_{x^2-y^2}$ band from the 10-band calculation at 0 GPa. The orbital is C_2 symmetric.

The data is currently uploaded on both NOMAD and TU Wien research data. In both instances the data was uploaded with embargo until publication. The referee should now be able to pre-view the data without institutional login at the address: <http://tinyurl.com/bdf2e36w>.

6. The extraction of Kanamori parameters from the Uijkl tensor from cRPA is not explained. Are these parameters extracted from the static limit ($\omega=0$) and then symmetrized? How well does the fit work?

The interaction matrices were calculated in Ref. [Phys. Rev. Lett. 124 (16), 166402] by the cRPA as implemented in VASP based on the approximations of Miyake *et al.* [Phys. Rev. B 80, 155134 (2009), Phys. Rev. 126, 413 (1962), Phys. Rev. 129, 62 (1963)] within the GGA in the framework of Perdew-Burke-Ernzerhof for solids version (GGA-PS), on a k -mesh of $13 \times 13 \times 15$.

A Wannier projection on the subspace of 5 Ni and 5 La or Pr Wannier functions was employed, and the static limit $U=U(\omega=0)$ was considered. From the U_{ijkl} matrix, the averaged Hubbard-Kanamori parameters were then computed using the formalism from Ref. [Phys. Rev. B 86, 165105]:

$$U = \frac{1}{N} \sum_{i=1}^{N=5} U_{iii}^{cubic}, \quad J = \frac{1}{N(N-1)} \sum_{i \neq j}^{N=5} U_{ijji}^{cubic}, \quad (1)$$

and $U' = U - 2J$. As for the quality of the Kanamori fit, the standard deviations of Eq. (1) are $\Delta U = 0.09$ eV and $\Delta J = 0.17$ eV for Pr and $\Delta U = 0.16$ eV and $\Delta J = 0.06$ eV for Ni.

Our cRPA interaction parameters are also consistent with other reports on $3d$ nickelates such as [Phys. Rev. B 85, 195102 (2012); Mater. Res. Express 3 (2016) 095701; Phys. Rev. B 84, 195450 (2011); and Phys. Rev. Lett. 107, 116805 (2011)], and $5d$ transition-metal oxides [Phys. Rev. Lett. 101, 076402 (2008); Phys. Rev. B 103, 165133 (2021); Phys. Rev. Lett. 125, 166402 (2020); Phys. Rev. B 90, 165105 (2014)].

7. I'm uncertain why the authors chose to use the Kanamori Hamiltonian in a full 5 orbital calculation for the wide energy DMFT calculations. This choice is inappropriate for a full 5 orbital d -shell due to the symmetry of the interaction. Why did the authors not opt for a Slater Hamiltonian instead? This could be a significant approximation, and the authors should at least acknowledge this fact.

We used the Kanamori Hamiltonian because it is numerically significantly more efficient and justified for fillings close to $3d^9$ with mostly filled t_{2g} -orbitals, see Fig. 4. Besides the roughly half-filled $3d_{x^2-y^2}$ orbital, the $3d_{z^2}$ has a small but noteworthy number of holes (due to its hybridization with the Pr states). However, the interaction between these two e_g orbitals is exactly of Kanamori form in the case of tetragonal symmetry.

We now comment on this in the Supplemental Material.

8. I'm unclear as to why the bare Coulomb interaction remains unchanged under pressure in the one orbital cRPA calculation. The band structures show a significant change in the bandwidth of the wannierized dx^2y^2 orbital. How does this not affect the bare Coulomb interaction? Is the spread of the Wannier functions constant under pressure? If so, do the authors have a theory as to why this is the case?

The referee's surmise is correct: In the current setup, the Wannier spread does not change perceptively with decreasing lattice constant. The reason for the seeming disparity between hopping elements (that grow notably under pressure) and the bare interaction (that decreases only slightly) lies foremost in the different spatial information they encode:

Figure 4: Occupation of the Ni (left) and Pr (right) d -orbitals under pressure at $x = 0$.

Electron transfer is a *non-local* process, described by a matrix element (of the Laplacian) between Wannier-functions of two neighboring sites. This “tunneling” of electrons from one site to the other depends exponentially on the distance between them (if we assume the shape and extent of the Wannier functions to be inert and interatomic distances to not be too small). Indeed, the orbital shape (which, in usual cases, does not evolve that much during the maximally localization procedure) is mainly a result of the orthogonalization with the lower-lying atomic orbitals of the same atom. For d -orbitals it typically necessitates pressures above 100 GPa to change.

A modification of the *local* Coulomb interaction, instead, *requires* a change of this shape/Wannier spread, as it is a matrix element involving four Wannier functions hosted by the same atomic site. For an illustrative model for Coulomb interactions under pressure, see Ref. [36].

9. The authors discuss the AFM order rather vaguely. As demonstrated by cluster DMFT studies (and other single site DMFT studies) found at <https://doi.org/10.1103/PhysRevB.105.205131>, the AFM order competes with superconductivity at around $\delta=0.12$ (filling of $d_{x^2-y^2}$ orbital) in these compounds. This competition could be due to the mean field nature of the calculations. Can the authors elaborate on how or if such a state could be observed in DGA? Is it correct to assume that the AFM order is essentially suppressed, or did the authors not observe any AFM order in their DMFT calculations? Is it known how the AFM phase evolves under pressure in simple one-band models and competes with the superconductivity?

Following the Reviewer’s comment, we looked into the DMFT susceptibility of our one-band model which indeed predicts antiferromagnetism without doping at lower pressures, see Fig. 5. With increasing pressure, antiferromagnetism (negative antiferromagnetic susceptibility within the enforced paramagnetic phase) goes away even in DMFT because of the increased self-doping. We hence do not consider competing antiferromagnetic order to be relevant at higher pressures.

Let us emphasize that DMFT *strongly overestimates* antiferromagnetism for (essentially) two-dimensional nickelates. DMFT antiferromagnetism will be suppressed by non-local fluctuations, as included in cluster DMFT (employed to nickelates in the above mentioned PRB) or DGA. It is also not found in experiment at zero or low pressures. The employed DGA suppresses antiferromagnetism completely, fulfilling the Mermin-Wagner theorem. We expect the actual antiferromagnetism to be somewhere in-between this DGA result and the (2×2) cluster DMFT, if a weak coupling in the third dimension and/or the finiteness of the experimental film is taken into account and leads to a finite Néel temperature.

This was indeed a very good point by the Reviewer and we now show the above Figure in the Supplemental material (Sect. IID3, Fig. S20) and briefly discuss it in the main text as well.

Figure 5: Inverse antiferromagnetic DMFT susceptibility $\chi_m(\pi, \pi, i\omega = 0)^{-1}$ without λ correction as a function of pressure and $x = 0$ at $\beta = 60$ in units of t (with t as reported in Tab. 1 of the main text). The pressure at which the AFM instability disappears even in DMFT is shown as a red x .

REVIEWERS' COMMENTS

Reviewer #2 (Remarks to the Author):

Below are my comments on the authors' response to my previous comments on the paper "Unconventional superconductivity without doping: infinite-layer nickelates under pressure":

In their paper (Main and Supplementary Information), and in their response, the authors have gone to great length to substantiate their claim that their theoretical pressure set-up is different from that in Ref 32. As they state, this is under the assumption that the STO in-plane lattice constants under pressure faithfully gives the in-plane lattice constants for the nickelate. I thank the authors for reiterating and clarifying this point; this will be helpful to the careful reader.

The authors now elaborate further on the Moriya correction; this is appreciated. However, since fit parameters are involved, the part of the work determining superconducting T_c (vis a vis $D\Gamma_{A}$) is not quite ab initio. And, as the authors agree, it is also not self-consistent, a task that is understandably computationally difficult. Following suggestion, they now state this more clearly in the Methods section and in Supplementary Information IG, IIB, IID, so readers can be free to judge the degree of accuracy of their predicted values of T_c 's.

The authors have now clarified in Sec IIB, the point regarding any possible divergence of the vertex, and the leading eigenvalue λ .

To address the question regarding calculations at a single temperature ($T=300K$), the authors have now presented calculations of relevant quantities at $T=100K$ at 50 GPa. That the authors took the time to do these extra calculations and present the results is appreciated, and will serve to reassure readers.

On the issue regarding self-consistency between their DFT and DMFT cycles, as suggested, the authors have now pointed out that their calculation is a single shot one. However, it is not clear that the effect of self-consistency would be as minor as claimed. The difference between the uncorrelated and correlated $Ni-d_x^2-d_y^2$ occupancy is found to be about 20%, which is not small. Absent a self-consistent DFT + DMFT calculation, it is not clear what the converged value of the occupancy would be. So, it would be more appropriate to remove the following: a) part of the sentence in Methods section that state "but the small change in orbital occupation from DFT to DMFT suggest that its effect is minor"; b) part of the sentence in Supplementary Information IF that state "because the changes in occupation were minor"; c) the last sentence in Supplementary Information IID.1 stating "Given the small difference in occupations we expect the effect of self-consistency between DFT and DMFT to be minor".

In response to my previous comment regarding the "incremental" nature of their work, the authors have emphasized that though they reproduce the increase of superconducting T_c under pressure for doped systems, their key new result is the prediction of superconductivity of the undoped compound at high pressure. I point out that my prior comments were not so much based on this work using the same technique (DFT + DMFT + $D\Gamma_{A}$) as previously used, but more so on the premise that the idea of exploring doping and pressure as dials for superconductivity in nickelate

was not new and had been previously explored (this notwithstanding the point regarding their pressure set-up being more realistic than that in other work (e.g. Ref 32)). And that not enforcing self-consistency between DFT and DMFT charges (a task that is feasible), together with not considering feedbacks between the particle-particle and particle-hole channels (a task that is understandably not so feasible) in the DGammaA calculation may reduce the robustness of the prediction of T_c .

In the end, I commend the authors for their efforts to address in detail all the questions posed, supplementing their original manuscript with additional calculations and presentation. Nevertheless, owing to reasons stated above, I have lingering reservations on the robustness of the quantitative results for T_c . But I concur with the authors' position that the accuracy of the results will have to be eventually determined by experiments.

I recommend publication of this paper, subject to the authors making the modifications suggested above. I leave it up to the Editors to decide on the specific Nature journal to publish the paper in.

Reviewer #3 (Remarks to the Author):

I thank the authors for their detailed answers to my questions. They answered all my questions adequately, explained open questions in great detail, and incorporated answers in the manuscript / supp. I am also very happy to see that some of my comments improved the quality / understanding of the manuscript. In my view, the prediction / finding of superconductivity under pressure without doping is a noteworthy result and I would recommend the manuscript in nature comm as is now.

Reviewer #3 (Remarks on code availability):

I took a quick look at the code and find it in good shape. The code seems to be well documented (for a research code anyway). I did not test the code myself, but readme and installation instructions seem to be easy to follow.

We thank both reviewers for their second round of comments. We are glad to see that reviewer 2 appreciated our additional effort, and for his/her/their fair review. We thank reviewer 3 for taking the time to check on our code.

Considering his remark:

On the issue regarding self-consistency between their DFT and DMFT cycles, as suggested, the authors have now pointed out that their calculation is a single shot one. However, it is not clear that the effect of self-consistency would be as minor as claimed. The difference between the uncorrelated and correlated Ni- $d_x^2-d_y^2$ occupancy is found to be about 20%, which is not small. Absent a self-consistent DFT + DMFT calculation, it is not clear what the converged value of the occupancy would be. So, it would be more appropriate to remove the following: a) part of the sentence in Methods section that state “but the small change in orbital occupation from DFT to DMFT suggest that its effect is minor”; b) part of the sentence in Supplementary Information IF that state “because the changes in occupation were minor”; c) the last sentence in Supplementary Information IID.1 stating “Given the small difference in occupations we expect the effect of self-consistency between DFT and DMFT to be minor”.

We have accepted this correction and changed main and supplementary accordingly.

Yours sincerely,

Simone Di Cataldo, Paul Worm, Liang Si, and Karsten Held